# The efficacy of current treatment processes to remove, inactivate, or reduce environmental bloom-forming *Escherichia coli*

Melody Lau,[1] Paul T. Monis,[1] Brendon J. King[1]

**ABSTRACT** *Escherichia coli* is excreted in high numbers from the intestinal tract of humans, other mammals, and birds. Traditionally, it had been thought that *E. coli* could grow only within human or animal hosts and would perish in the environment. Therefore, the presence of *E. coli* in water has become universally accepted as a key water quality indicator of fecal pollution. However, recent research challenges the assumption that the presence of *E. coli* in water is always an indicator of fecal contamination, with some types of *E. coli* having evolved to survive and grow in aquatic environments. These strains can form blooms in water storages, resulting in high *E. coli* counts even without fecal contamination. Although these bloom-forming strains lack virulence genes and pose little threat to public health, their presence in treated water triggers the same response as fecal-derived *E. coli*. Yet, little is known about the effectiveness of treatment processes in removing or inactivating them. This study evaluated the effectiveness of current treatment processes to remove bloom-forming strains, in comparison to fecal-derived strains, with conventional coagulation-flocculation-sedimentation and filtration investigated. Second, the effectiveness of current disinfection processes—chlorination, chloramination, and ultraviolet (UV) light to disinfect bloom-forming strains in comparison to fecal-derived strains—was assessed. These experiments showed that the responses of bloom isolates were not significantly different from those of fecal *E. coli* strains. Therefore, commonly used water treatment and disinfection processes are effective to remove bloom-forming *E. coli* strains from water.

**IMPORTANCE** The presence of *Escherichia coli* in water has long been used globally as a key indicator of fecal pollution and for quantifying water safety. Traditionally, it was believed that *E. coli* could only thrive within hosts and would perish outside, making its presence in water indicative of fecal contamination. However, recent research has unveiled strains of *E. coli* capable of surviving and proliferating in aquatic environments, forming blooms even in the absence of fecal contamination. While these bloom-forming strains lack the genes to be pathogenic, their detection in source or drinking water triggers the same response as fecal-derived *E. coli*. Yet, little is known about the efficacy of treatment processes in removing them. This study evaluated the effectiveness of conventional treatment and disinfection processes in removing bloom-forming strains compared to fecal-derived strains. Results indicate that these commonly used processes are equally effective against both types of *E. coli*, reassuring that bloom-forming *E. coli* strains can be eliminated from water.

**KEYWORDS** *Escherichia coli*, bloom-forming, treatment, disinfection

*E*scherichia coli is used as a key water quality indicator for drinking water treatment and distribution systems worldwide (1, 2). In addition to providing utilities with

Address correspondence to Brendon J. King, brendon.king@sawater.com.au.

The authors declare no conflict of interest.

See the funding table on p. 18.

valuable information on process efficiency, both the World Health Organisation (WHO) and the United States Environmental Protection Agency (USEPA) guidelines recommend *E. coli* as an indicator of fecal contamination (3, 4). In an Australian context, the implementation of health-based targets further increases the reliance on the use of *E. coli* for determining the treatment requirements for source waters (5).

In the absence of fecal contamination, an ideal fecal indicator—among other characteristics—should not be present in a water body or be capable of multiplying outside a host (2, 6–14). However, research undertaken over the last two decades, challenges the assumption that the presence of *E. coli* in water is always an indicator of fecal contamination (2, 9–12, 14, 15). In particular, some types of *E. coli* have evolved to survive and grow in aquatic environments (16–18). Under favorable conditions, these strains can form blooms in raw water storages—resulting in extremely high *E. coli* counts even in the absence of fecal contamination (9, 10, 15). While these bloom-forming *E. coli* lack the genes required to cause disease in humans (16), and therefore are unlikely to be a threat to public health, little is known regarding the efficacy of treatment barriers to remove or inactivate these microorganisms. This is especially important as their presence in treated water can trigger the same operational response as the detection of fecal-derived *E. coli* (19).

Current knowledge regarding the removal and inactivation of *E. coli* by drinking water processes to date has been based on fecal isolates (20–23). Presently, the detection of *E. coli* at the outlet of a drinking water treatment plant following disinfection would be interpreted as a treatment failure, with the detection of high numbers of *E. coli* in the product water potentially triggering a boiled water advisory. Considering that most bloom-forming *E. coli* have additional protection in the form of mucoid capsular material of *Klebsiella* origin (9, 15) shown to confer a growth advantage (13), it is possible that they may respond differently to water treatment and disinfection processes than fecal *E. coli*, which lack this additional protective layer.

The aim of this work was to evaluate the effectiveness of current treatment processes in removing bloom-forming strains in comparison to fecal-derived strains. First, we sought to evaluate conventional coagulation-flocculation-sedimentation and filtration processes. Second, we sought to assess the effectiveness of current disinfection processes—chlorination, chloramination, and ultraviolet (UV) light—in disinfecting bloom-forming strains compared to fecal-derived strains. These experiments showed that the responses of bloom isolates used in this study were not significantly different from those of fecal *E. coli* strains. Therefore, commonly used water treatment and disinfection processes should be considered effective for removing bloom-forming *E. coli* strains from water.

## RESULTS

### Water treatment and disinfection methods

The effectiveness of water treatment and disinfection processes is measured using a concept called "log removal values" (LRVs). LRVs are determined by the logarithm of the ratio of pathogen concentration in the influent (before treatment) and effluent (after treatment) of a treatment process. For example, an LRV of 1 is equivalent to 90% removal of target pathogen; an LRV of 2 is equivalent to 99% removal, while an LRV of 3 is equivalent to 99.9% removal, and so forth. Treatment and disinfection removals are, therefore, presented throughout the results section as LRVs.

### Jar testing to mimic full-scale Happy Valley water treatment plant

The coagulant used at Happy Valley treatment plant is Aluminum sulfate ($Al_2(SO_4)$ $\cdot 18H_2O$, alum), and alum was chosen for jar testing of the raw water sourced from this location. Figure 1 shows the LRVs of coagulation-flocculation-sedimentation and filtration of six "starved" *E. coli* strains under optimal alum dosing. Data collected during jar testing indicated that for the treatment processes tested, no statistically significant

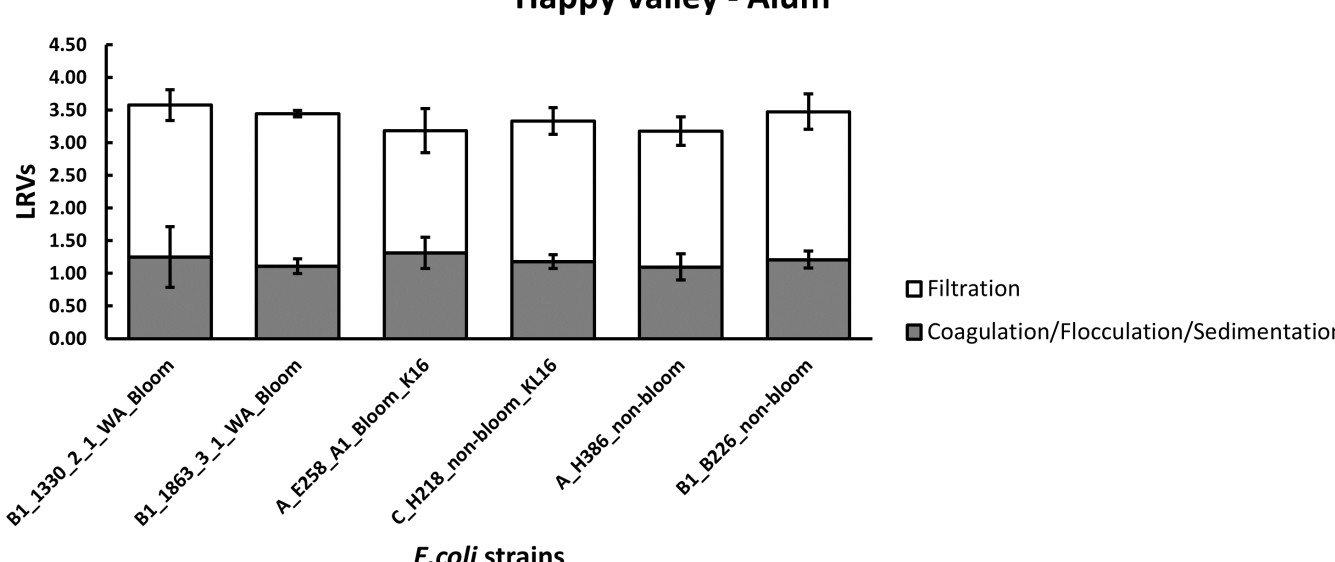

**FIG 1** Average LRVs of "starved" *E. coli* following coagulation-flocculation-sedimentation and filtration of Happy Valley Reservoir water. Data illustrate the mean and standard deviation from three replicate experiments.

response to coagulation or filtration was observed between any of the *E. coli* strains. LRVs for coagulation of all six "starved" *E. coli* strains were similar between all types (i.e., bloom vs non-bloom formers and encapsulated vs non-encapsulated), ranging from 1.11 to 1.31 log10. Similar removal effectiveness of "starved" *E. coli* for filtration was observed across all strains tested, ranging from 1.87 to 2.34 log10. This resulted in a combined (coagulation-flocculation-sedimentation and filtration) removal of approximately 3.18 to 3.57 log10 across all six *E. coli* strains. Regardless of how strains were grouped, no statistically significant differences in removals were apparent in response to the treatment processes tested.

Water quality parameters of Happy Valley water used in jar testing (before and after treatment) are presented within Table S1. Water quality did not change over the course of the experimental period. At the beginning of experimentation, turbidity, color, UV absorbance, and DOC were 12.8 NTU, 47 HU, 0.29 mg/L, and 8.2 mg/L, respectively. The removals of DOC, UV absorbance, color, and turbidity were similar throughout all jar testing for each of the six strains tested: with average removals for turbidity, color, UV absorbance, and DOC of 81%, 87%, 71%, and 47%, respectively.

## Jar testing to mimic full-scale Prospect Water treatment plant

The coagulant used at Prospect treatment plant is ferric chloride ($FeCl_3$) and was chosen for jar testing of the raw water sourced from this location. Figure 2 shows the LRVs of coagulation-flocculation-sedimentation and filtration of six "starved" *E. coli* strains under optimal ferric chloride dosing. The jar testing data indicated that the LRVs for coagulation-flocculation-sedimentation treatment of all six "starved" *E. coli* strains were highly similar, ranging from 0.46 to 0.66 log10. No statistically significant differences in coagulation-flocculation-sedimentation behavior were observed across any of the strains tested. Additionally, similar removal effectiveness of "starved" *E. coli* after filtration was observed for all strains tested, ranging from 3.12 to 3.37 log10, with the exception of A_E258 strain, where only a 2.46 log10 removal was achieved after filtration. This strain (A_E258) exhibited significantly poorer filtration removal compared to other strains tested (*P* value < 0.05), with the exception of C_H218_non-bloom_KL16. Therefore, a combined (coagulation-flocculation-sedimentation and filtration) removal of approximately 3.7 to 3.9 log10 was achieved across all *E. coli* strains, with the exception of

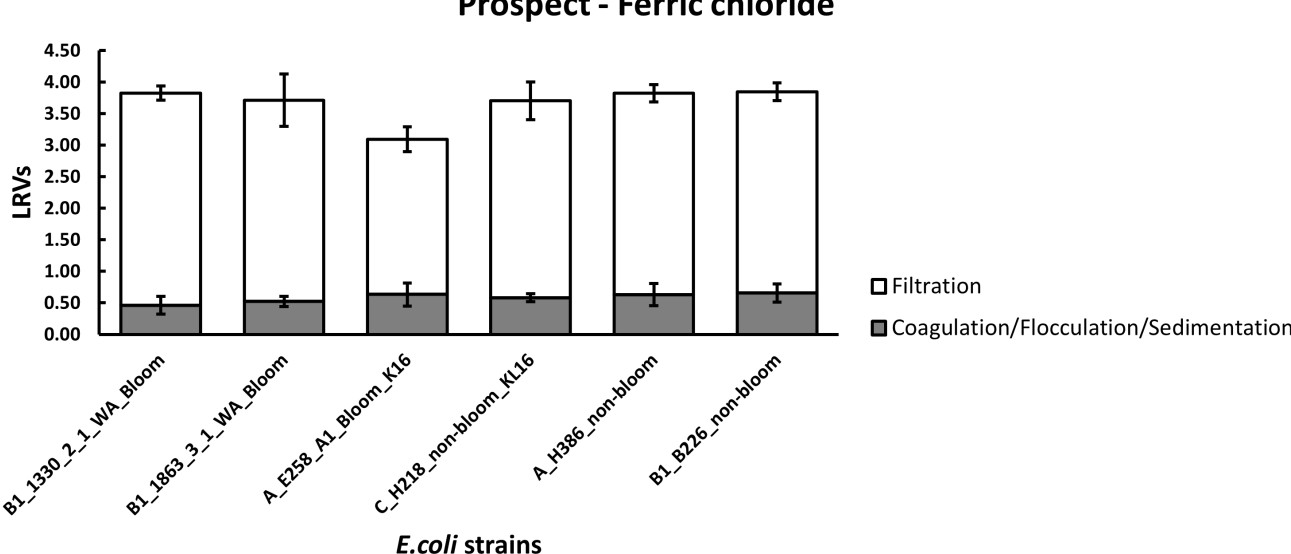

**FIG 2** Average LRVs of "starved" *E. coli* during coagulation-flocculation-sedimentation and filtration of Prospect Reservoir water. Data illustrate the mean and standard deviation from three replicate experiments.

A_E258_A1_bloom_K16 strain, where only a 3.09 log10 was achieved due to poorer filtration removal characteristics. However, when strains were grouped either as bloom vs non-bloom formers or capsule vs non-capsule *E. coli*, no statistically significant differences in removals to the treatment processes investigated were apparent. Water quality parameters of Prospect water used in jar testing (before and after treatment) are presented within Table S2. Water quality did not change over the course of the experimental period. At the beginning of experimentation, turbidity, color, UV absorbance, and DOC were 1.93 NTU, 7 HU, 0.10 mg/L and 4.6 mg/L, respectively. The removals of DOC, UV absorbance, color, and turbidity were similar throughout all jar testing for each of the six strains tested, with average removals for turbidity, color, UV absorbance, and DOC of 16%, 21%, 15% and 6%, respectively.

### Chlorine disinfection of *E. coli*

Table 1 shows the calculated mean CT values for 1, 2, and 3 LRVs for "fresh" *E. coli* after chlorination across a range of temperatures and pH conditions. The CT data indicated *E. coli* was able to be effectively disinfected in demand-free water at low chlorine doses, with CTs as low as 0.044, 0.091, and 0.139 mg/L·min able to achieve corresponding LRVs of 1, 2, and 3 for the WA Bloom strain B1_1330, at pH 7.5, 25°C. The CT values for all six "fresh" *E. coli* strains were highly similar between strain types (i.e., bloom vs non-bloom formers, capsule vs non-encapsulated and different phylogroups). No statistically significant differences in CT values between each of the six strains were observed among the 3 pHs (7, 7.5, and 8) or temperature (10°C and 25°C) regimes tested, with minimal variation between experimental replicates observed.

Table 2 shows the CT values for "starved" *E. coli* after chlorination at pH 7.5, 25°C. For the strains examined, the CT values of "starved" *E. coli* were similar to that of "fresh" *E. coli* (not significantly different) under this pH and temperature regime, indicating no difference in disinfection due to the metabolic nature of *E. coli* ("starvation" vs "fresh").

### *In situ* monochloramine disinfection of *E. coli*

Table 3 shows the calculated mean CT values for 1, 2, and 3 LRVs for "fresh" *E. coli* after *in situ* monochloramine disinfection across a range of temperatures and pH conditions.

**TABLE 1** CT values for "fresh" *E. coli* inactivation using chlorination[a]

| Disinfectant | pH | Temp | B1_1330_2_1_WA_Bloom | B1_1863_3_1_WA_Bloom | A_E258_A1_Bloom_K16 | C_H218_non bloom_KL16 | A_H386 non-bloom | B1_B226_non-bloom |
|---|---|---|---|---|---|---|---|---|
| CT values for 90% inactivation | | | | | | | | |
| Free chlorine | 7.5 | 25 | 0.044 ± 0.004 | 0.065 ± 0.019 | 0.062 ± 0.022 | 0.055 ± 0.017 | 0.062 ± 0.021 | 0.064 ± 0.019 |
| | 7.5 | 10 | 0.054 ± 0.001 | 0.070 ± 0.016 | 0.066 ± 0.007 | 0.055 ± 0.011 | 0.077 ± 0.008 | 0.060 ± 0.014 |
| | 7 | 25 | 0.063 ± 0.014 | 0.067 ± 0.011 | 0.054 ± 0.014 | 0.070 ± 0.018 | 0.065 ± 0.018 | 0.060 ± 0.005 |
| | 8 | 25 | 0.061 ± 0.010 | 0.061 ± 0.017 | 0.073 ± 0.011 | 0.072 ± 0.019 | 0.078 ± 0.012 | 0.077 ± 0.014 |
| CT values for 99% inactivation | | | | | | | | |
| Free chlorine | 7.5 | 25 | 0.091 ± 0.010 | 0.153 ± 0.060 | 0.153 ± 0.073 | 0.140 ± 0.063 | 0.160 ± 0.048 | 0.154 ± 0.068 |
| | 7.5 | 10 | 0.108 ± 0.001 | 0.139 ± 0.031 | 0.137 ± 0.015 | 0.111 ± 0.021 | 0.155 ± 0.016 | 0.119 ± 0.029 |
| | 7 | 25 | 0.125 ± 0.029 | 0.133 ± 0.023 | 0.109 ± 0.029 | 0.140 ± 0.036 | 0.130 ± 0.036 | 0.121 ± 0.011 |
| | 8 | 25 | 0.121 ± 0.019 | 0.123 ± 0.033 | 0.145 ± 0.022 | 0.145 ± 0.037 | 0.156 ± 0.023 | 0.154 ± 0.028 |
| CT values for 99.9% inactivation | | | | | | | | |
| Free chlorine | 7.5 | 25 | 0.139 ± 0.021 | 0.230 ± 0.089 | 0.229 ± 0.109 | 0.210 ± 0.095 | 0.240 ± 0.072 | 0.231 ± 0.103 |
| | 7.5 | 10 | 0.162 ± 0.002 | 0.209 ± 0.047 | 0.206 ± 0.022 | 0.166 ± 0.032 | 0.232 ± 0.025 | 0.179 ± 0.043 |
| | 7 | 25 | 0.188 ± 0.043 | 0.200 ± 0.034 | 0.163 ± 0.043 | 0.209 ± 0.054 | 0.194 ± 0.055 | 0.181 ± 0.016 |
| | 8 | 25 | 0.182 ± 0.029 | 0.184 ± 0.050 | 0.218 ± 0.033 | 0.217 ± 0.056 | 0.233 ± 0.035 | 0.231 ± 0.042 |

[a]Data show the mean and standard deviation from three replicate experiments.

**TABLE 2** CT values for "starved" *E. coli* using chlorination at pH 7.5, 25°C[a]

| LRV | B1_1330_2_1_WA_Bloom | B1_1863_3_1_WA_Bloom | A_E258_A1_Bloom_K16 | C_H218_non-bloom_KL16 | A_H386 non-bloom | B1_B226_non-bloom |
|---|---|---|---|---|---|---|
| 1 | 0.053 ± 0.019 | 0.053 ± 0.017 | 0.050 ± 0.001 | 0.045 ± 0.008 | 0.058 ± 0.015 | 0.043 ± 0.006 |
| 2 | 0.091 ± 0.010 | 0.111 ± 0.040 | 0.106 ± 0.002 | 0.096 ± 0.019 | 0.124 ± 0.032 | 0.091 ± 0.014 |
| 3 | 0.159 ± 0.057 | 0.178 ± 0.071 | 0.172 ± 0.007 | 0.158 ± 0.035 | 0.202 ± 0.030 | 0.146 ± 0.026 |

[a]Data show the mean and standard deviation from three replicate experiments.

CT values for all six "fresh" *E. coli* strains were similar between all strain types (i.e., bloom vs non-bloom formers, capsule vs non-encapsulated and different phylogroups). No statistically significant differences in CT values for "fresh" *E. coli* were observed over the temperature regimes examined (10°C and 25°C, at pH 7.5). For those pH conditions investigated, no significant difference in disinfection behavior was observed across strains, with the exception of the B1_1863_3_1_WA_Bloom strain, which was substantially more resistant at pH 8.0 (*P* value < 0.05). For instance, the CT value to achieve a 1 LRV of the WA Bloom strain B1 1863 at pH 8, 25°C (0.542 mg/L·min) was seven times higher when compared to the CT value at pH7.5 (0.070 mg/L·min). Overall, no significant differences in disinfection behavior between bloom vs non-bloom formers or capsule vs non-capsule *E. coli* strains were apparent for the conditions tested.

Of special note, the CT values calculated for the *in situ* chloramination were similar to chlorination at pH 7.0 and 7.5, while at pH 8.0, the CTs for *in situ* chloramination were consistently greater than those CTs for chlorination, and substantially more variable. Additionally, it was observed that a low level of free chlorine (<0.1 mg/L) was present throughout these chloramination experiments. These factors in combination with the lower CTs required achieving a high degree of *E. coli* inactivation—compared to that found in the literature—were, therefore, attributed to exposure to free available chlorine during monochloramine formation. Table 4 shows the CT values of "starved" *E. coli* for chloramination at pH 7.5 at 25°C. The CT values of "starved" *E. coli* were similar to "fresh" *E. coli*, suggesting there was no difference in inactivation of *E. coli* after starvation under this pH and temperature regime. Based on the above results, it was decided that preformed monochloramine disinfection experiments needed to be undertaken to compare the different strains as the disinfection witnessed in the *in situ* experiments was primarily the result of chlorine and not monochloramine.

### Preformed monochloramine disinfection of *E. coli*

From the *in situ* monochloramination disinfection experiments, it was determined that the *E. coli* disinfection measured was predominately the result of the free chlorine disinfection at the lower pHs tested and not as a result of monochloramine disinfection. As monochloramine is predominantly formed at higher pH, it was decided to repeat the *E. coli* disinfection experiments with preformed monochloramine at pH 8.5. Table 5 shows the calculated mean CT values of "starved" *E. coli* for the preformed monochloramine at pH 8.5, 25°C. The CT values indicated that inactivation was similar across bloom formers (B1_1330_2_1_WA_Bloom, B1_1863_3_1_WA_Bloom, and E258_A1_bloom_K16), with the CT values for 1, 2, 3 LRVs ranging from 5.21 to 6.79, 9.60–10.80, 13.59–14.09 mg/L·min, respectively. Disinfection of non-bloomers (C_H218_non-bloom_KL16, A_H386 non-bloom and B1_B226_non-bloom) produced more variable results. The CT values for 1, 2, 3 LRVs ranged from 3.24 to 7.89, 6.81 to 12.86, and 10.85 to 17.20 mg/L·min, respectively. However, no significant difference in disinfection behavior was observed between bloom formers vs non-bloom formers or capsule vs non-capsule *E. coli* strains. Inactivation of *E. coli* with preformed monochloramine was substantially slower than *in situ* monochloramine (*P* value < 0.05).

**TABLE 3** CT values for inactivation of "fresh" *E. coli* using *in situ* monochloramination[a]

| Disinfectant | pH | Temp | B1_1330_2_1_WA_Bloom | B1_1863_3_1_WA_Bloom | A_E258_A1_Bloom_K16 | C_H218_non bloom_KL16 | A_H386 non-bloom | B1_B226_non-bloom |
|---|---|---|---|---|---|---|---|---|
| CT values for 90% inactivation | | | | | | | | |
| *In situ* monochloramine | 7.5 | 25 | 0.061 ± 0.019 | 0.070 ± 0.031 | 0.077 ± 0.025 | 0.032 ± 0.023 | 0.068 ± 0.017 | 0.066 ± 0.015 |
| | 7.5 | 10 | 0.050 ± 0.004 | 0.056 ± 0.015 | 0.097 ± 0.049 | 0.057 ± 0.021 | 0.057 ± 0.007 | 0.125 ± 0.082 |
| | 7 | 25 | 0.062 ± 0.013 | 0.080 ± 0.018 | 0.063 ± 0.010 | 0.085 ± 0.013 | 0.072 ± 0.017 | 0.071 ± 0.019 |
| | 8 | 25 | 0.077 ± 0.004 | 0.542 ± 0.044 | 0.077 ± 0.012 | 0.140 ± 0.099 | 0.175 ± 0.095 | 0.294 ± 0.189 |
| CT values for 99% inactivation | | | | | | | | |
| *In situ* monochloramine | 7.5 | 25 | 0.126 ± 0.033 | 0.141 ± 0.062 | 0.153 ± 0.049 | 0.063 ± 0.047 | 0.135 ± 0.035 | 0.132 ± 0.030 |
| | 7.5 | 10 | 0.100 ± 0.009 | 0.112 ± 0.030 | 0.171 ± 0.057 | 0.115 ± 0.042 | 0.114 ± 0.013 | 0.231 ± 0.131 |
| | 7 | 25 | 0.127 ± 0.032 | 0.161 ± 0.035 | 0.126 ± 0.021 | 0.171 ± 0.027 | 0.143 ± 0.034 | 0.146 ± 0.044 |
| | 8 | 25 | 0.153 ± 0.008 | 0.787 ± 0.640 | 0.154 ± 0.025 | 0.281 ± 0.198 | 0.303 ± 0.154 | 0.588 ± 0.377 |
| CT values for 99.9% inactivation | | | | | | | | |
| *In situ* monochloramine | 7.5 | 25 | 0.199 ± 0.036 | 0.211 ± 0.093 | 0.230 ± 0.074 | 0.095 ± 0.070 | 0.203 ± 0.052 | 0.198 ± 0.045 |
| | 7.5 | 10 | 0.151 ± 0.013 | 0.168 ± 0.045 | 0.238 ± 0.054 | 0.172 ± 0.062 | 0.171 ± 0.020 | 0.328 ± 0.164 |
| | 7 | 25 | 0.199 ± 0.062 | 0.241 ± 0.053 | 0.188 ± 0.031 | 0.256 ± 0.040 | 0.215 ± 0.050 | 0.228 ± 0.080 |
| | 8 | 25 | 0.230 ± 0.011 | 1.027 ± 0.758 | 0.232 ± 0.037 | 0.421 ± 0.297 | 0.413 ± 0.196 | 0.883 ± 0.565 |

[a]Data show the mean and standard deviation from three replicate experiments.

**TABLE 4** CT values of "starved" *E. coli* inactivation using *in situ* monochloramination at pH 7.5, 25°C

| LRV | B1_1330_2_1_ WA_Bloom | B1_1863_3_1_ WA_Bloom | A_E258_A1_ Bloom_K16 | C_H218_non- bloom_KL16 | A_H386 non-bloom | B1_B226_non-bloom |
|---|---|---|---|---|---|---|
| 1 | 0.073 ± 0.028[a] | 0.053 ± 0.004 | 0.066 ± 0.020 | 0.046 ± 0.047 | 0.082 ± 0.015 | 0.071 ± 0.018 |
| 2 | 0.145 ± 0.056 | 0.106 ± 0.007 | 0.132 ± 0.039 | 0.092 ± 0.094 | 0.168 ± 0.033 | 0.152 ± 0.036 |
| 3 | 0.218 ± 0.084 | 0.159 ± 0.011 | 0.198 ± 0.059 | 0.137 ± 0.141 | 0.259 ± 0.058 | 0.213 ± 0.054 |

[a]Data show the mean and standard deviation from three replicate experiments.

## UV disinfection of *E. coli*

Tables 6 and 7 show the UV doses required to achieve 1, 2, and 3 LRVs for "fresh" and "starved" *E. coli,* respectively, for all six strains investigated. The inactivation results achieved throughout the UV inactivation experiments conducted herein were largely in line with the ranges reported by the US EPA (24) (Table 6). Some of the six strains tested exhibited significant differences in UV disinfection; however, no significant differences were apparent between groups comprising strains representing bloom vs non-bloom formers or encapsulated vs non-capsule.

For those bloom formers examined in this study, the required UV dose to achieve a 1, 2, 3 LRV were similar for *E. coli* strains belonging to the B1 phylogroup (B1_1863_3_1_WA_bloom and B1_1330_2_1_WA_bloom), while the A phylogroup (A_E258_bloom_K16) required a significantly higher UV dose to achieve the same LRVs (*P* value < 0.001). The presence or absence of a capsule within the phylogroup B1 (B1_1863_3_1_WA_bloom and B1_1330_2_1_WA_bloom) had no significant effect on UV disinfection. The bloom-forming strain A_E258_bloom_K16, which belongs to phylogroup A, had LRVs that were significantly different to all other strains tested, suggesting that A_E258 was the most resistant *E. coli* strain tested for UV disinfection (*P* values ranging from <0.05 to 0.001). For those non-bloom formers examined in this study, the required UV dose for inactivation of encapsulated *E. coli* (C_H218_non-bloom_KL16) was significantly lower than non-encapsulated (A_H386 non-bloom and B1_B226_non-bloom, *P* < 0.01). There was no significant difference in phylogroup A (A_H386 non-bloom) and phylogroup B1 (B1_B226_non-bloom) among non-bloom formers. The UV dose required for inactivation of "fresh" and "starved" *E. coli* was similar. There was no significant difference in disinfection behavior, suggesting no change in inactivation of *E. coli* after starvation.

## DISCUSSION

*E. coli* has been used universally as the preferred indicator of fecal contamination, not only in drinking water, but also in other matrices (25). Additionally, it is employed as a general indicator to determine the performance of water treatment and disinfection processes (26, 27). However, the presence of environmental (free-living) and naturally occurring bloom-forming *E. coli* challenges the assumption that the presence of *E. coli* in water is always an indicator of fecal contamination (9, 11, 12, 28–30).

Bloom formers—which are not human pathogens—encompass strains that typically carry genes for the synthesis of a group 1 capsule, similar to that found in *Klebsiella* (9, 15). It has been hypothesized that the presence of this capsule may be critical for bloom formation—enhancing survival within the environment by providing protection from adverse conditions, including ultraviolet (UV) radiation, desiccation, osmotic stress,

**TABLE 5** CT values of "starved" *E. coli* for preformed monochloramine at pH 8.5, 25°C

| LRV | B1_1330_2_1 _WA_Bloom | B1_1863_3_1 _WA_Bloom | A_E258_A1_ Bloom_K16 | C_H218_non- bloom_KL16 | A_H386 non-bloom | B1_B226_non- bloom |
|---|---|---|---|---|---|---|
| 1 | 6.79 ± 2.24[a] | 5.21 ± 1.35 | 5.27 ± 2.07 | 3.24 ± 0.58 | 5.33 ± 1.79 | 7.89 ± 3.20 |
| 2 | 10.80 ± 2.38 | 9.60 ± 1.60 | 9.62 ± 2.68 | 6.81 ± 1.08 | 9.39 ± 1.97 | 12.86 ± 3.36 |
| 3 | 14.09 ± 2.33 | 13.59 ± 1.50 | 13.61 ± 2.72 | 10.85 ± 1.39 | 12.97 ± 1.80 | 17.20 ± 2.88 |

[a]Data show the mean and standard deviation from three replicate experiments.

**TABLE 6** UV doses (mJ/cm$^2$) required for inactivation of "fresh" *E. coli*

| LRV | *E. coli*, US EPA (2003) | B1_1330_2_1_WA_Bloom | B1_1863_3_1_WA_Bloom | A_E258_A1_Bloom_K16 | C_H218_non-bloom_KL16 | A_H386 non-bloom | B1_B226_non-bloom |
|---|---|---|---|---|---|---|---|
| 1 | 1.5–4.4 | 1.37 ± 0.39 | 1.02 ± 0.05[a] | 3.48 ± 1.10 | 0.56 ± 0.04 | 1.67 ± 0.35 | 2.23 ± 0.30 |
| 2 | 2.8–6.2 | 2.40 ± 0.30 | 1.81 ± 0.09[a] | 5.54 ± 0.85 | 1.18 ± 0.08 | 3.11 ± 0.45 | 3.84 ± 0.36 |
| 3 | 4.1–7.3 | 3.32 ± 0.13 | 2.47 ± 0.12[a] | 7.27 ± 0.70 | 1.87 ± 0.13 | 4.44 ± 0.45 | 5.19 ± 0.49 |

[a]Data show the mean and standard deviation from three replicate experiments.

predation by protozoa, and bacteriophage infection (31, 32). Furthermore, recent work has demonstrated that the capsule may facilitate nutrient uptake or utilization providing a growth rate advantage compared to nonencapsulated strains (13). This raises the specter that bloom-forming *E. coli*, which have additional protection in the form of mucoid capsular material (9), may also behave differently in water treatment and disinfection processes (33). Understanding the behavior of bloom-forming *E. coli* in common water treatment processes is crucial as their presence in catchments or product water may incorrectly suggest a significant fecal contamination event; inadvertently triggering a major water quality incident. This study aimed to examine the effectiveness of current treatment processes on several *E. coli* strains ("fresh" and "starved") encompassing both bloom and non-bloom formers, and encapsulated vs non-encapsulated strains.

Coagulation followed by flocculation as an aid to sedimentation and filtration are commonly used treatment processes. Coagulant and flocculant-aid chemicals are used to remove particles, some natural organic matter, color and microorganisms by aggregating fine suspended matter into larger flocs (Coagulation-Flocculation). When the flocs reach a certain size, they settle, allowing the clarified water to be more effectively filtered (filtration). The performance of water treatment processes is defined by log removal values (LRVs). In an Australian context, the default LRV treatment credit for coagulation-flocculation-sedimentation and filtration for removal of bacteria quoted by the Health Based Target Manual (5) is 1 log10 each, with a total value of 2 log10 for the combined processes.

Jar testing of Happy Valley water with alum demonstrated coagulation-flocculation-sedimentation LRVs for bacteria removal highly similar to the default value of 1 log10 (1.11–1.31 log10). However, jar testing of Prospect water with ferric chloride demonstrated LRVs for bacteria removal lower than the default value (0.46–0.66 log10). These differences in removals may be a result of the substantial differences in source water quality (Tables S1 and S2) and treatment conditions (coagulant addition and pH). Ferric and aluminum-based coagulants, although operating by similar mechanisms (34), differ mostly by the pH range in which they best operate. Coagulants perform best at or near their isoelectric point (35). For aluminum-based coagulants, this is at or near pH 5.5, while for ferric coagulants, it is closer to pH 4.5. Coagulation conditions were closer to optimum conditions for Happy Valley water treatment compared with Prospect water, hence the higher LRVs observed for coagulation-flocculation-sedimentation of Happy Valley water. However, coagulation at pH less than 6 is unsustainable as it creates a corrosive environment for WTP infrastructure, so ferric coagulants are almost never applied near its optimum pH, and therefore, any performance benefit from the slightly higher electro-positivity of ferric coagulants is never realized. Importantly, the assessment of LRVs for coagulation-flocculation-sedimentation for both waters—Happy Valley

**TABLE 7** UV doses (mJ/cm$^2$) required for inactivation of "starved" *E. coli*[a]

| LRV | B1_1330_2_1_WA_bloom | B1_1863_3_1_WA_Bloom | A_E258_A1_Bloom_K16 | C_H218_non bloom_KL16 | A_H386 non-bloom | B1_B226_non-bloom |
|---|---|---|---|---|---|---|
| 1 | 1.24 ± 0.16 | 1.24 ± 0.16 | 3.82 ± 0.46 | 0.81 ± 0.23 | 1.57 ± 0.29 | 2.79 ± 0.22 |
| 2 | 2.46 ± 0.27 | 2.22 ± 0.11 | 5.73 ± 0.36 | 1.66 ± 0.43 | 2.97 ± 0.41 | 4.31 ± 0.17 |
| 3 | 3.67 ± 0.34 | 3.10 ± 0.10 | 7.21 ± 0.27 | 2.55 ± 0.61 | 4.26 ± 0.44 | 5.50 ± 0.14 |

[a]Data show the mean and standard deviation from three replicate experiments.

and Prospect—showed no significant differences in removal of bloom vs non-bloom formers or encapsulated vs non-encapsulated *E. coli* strains.

The combined coagulation-flocculation-sedimentation and filtration processes for investigations using Happy Valley water achieved LRVs for bacteria considerably greater than the default value of 2 log10 (3.18–3.57 log10). Additionally, investigations using Prospect source water also achieved substantially greater removals than the default value for the combined treatment process (3.7–3.9 log10) even though the LRVs for the coagulation-flocculation-sedimentation of this water were poorer. However, it should be noted that at Prospect Water Filtration Plant, filters are designed to enable solids to be removed directly on the filters without requiring a sedimentation stage. The greater LRVs for filtration of the Prospect water may be the result of some flocs not settling in the coagulation-flocculation process but that were then removed efficiently by filtration. These differences between the default values and LRVs calculated from laboratory experiments are not unexpected. Default values tend to be conservative, and since filtration can be significantly influenced by numerous factors including, but not limited to: medium type, grain size, presence of biofilm and natural organic material (NOM), biotic predation, ionic strength, and pH. Laboratory and pilot-scale experiments simulating conventional treatment plants with filters often demonstrate higher removals for micro-organisms (36, 37). Furthermore, operational parameters such as hydraulic load, chemical pre-treatments, and backwashing regimes can greatly affect removal efficiency. In this case, filtration was simulated using a Whatman No.1 membrane (nominal pore size 11 µm), enabling a uniform comparison of coagulated water treatments with various *E. coli* strains chosen for investigation. It is important to note that, by having significantly different characteristics to media filtration and operating parameters used at WTPs, this approach was limited in being able to compare *in vitro* obtained LRVs to those achieved in WTPs.

Aside from the differences evident in the treatment characteristics of the source waters examined, or difference in the filterability of an individual *E. coli* strain, no significant differences were apparent in the removal of bloom vs non-bloom formers or encapsulated vs non-encapsulated *E. coli* strains. The presence of bacterial capsules potentially favors the attachment to surfaces and the formation of biofilms through non-specific forces such as electrostatic and van der Waals forces and/or specific ligand-receptor interactions (38–40). Due to their considerable additional mucoid capsular protection layer, bloom and capsule positive strains could be envisaged to differentially interact with flocs or adherence to filters compared to non-encapsulated strains. However, this was not observed under the treatment conditions described and tested herein (i.e., coagulation/flocculation/ sedimentation, filtration, or the combined treatment processes). Nevertheless, it needs to be noted that due to logistical constraints, only a limited number of *E. coli* strains were investigated. A more extensive investigation between bloom/non-bloom or encapsulated/nonencapsulated types may identify differences in treatability between some members of these groups. Furthermore, in selecting strains for testing, the option of focusing on isolates derived from the detection of environmental *E. coli*, specifically those encapsulated forms within product waters could be considered.

The effectiveness of disinfection greatly depends on the water quality; thus, disinfection typically occurs after treatment processes (e.g., coagulation-flocculation-sedimentation and filtration) that clarify the source water—removing turbidity, color, organic carbon as well as microorganisms. The most commonly used disinfection processes are chlorination, chloramination, and ultraviolet irradiation (UV). The performance of both chlorination and chloramination is highly dependent on disinfectant concentration, contact time, temperature, pH, organic matter concentration, and water turbidity. Chlorination is widely used for drinking water disinfection due to its relatively low cost, ease of use, and low operational power requirements. Chlorine is a strong oxidizing agent that hydrolyzes in water to form hypochlorous acid (HOCl), which oxidises cell walls and leads to cell lysis or inactivation of functional sites on

the cell surface (41). The CT values obtained in this study were similar for both bloom and non-bloom formers as well as encapsulated and non-encapsulated *E. coli* strains, suggesting similar inactivation rates for chlorination disinfection. Additionally, CT values in this study were concordant with previously published data. For example, a 3.8–4 log10 disinfection for both low and high infectious (entero-haemorrhagic) *E. coli* strains was achieved with a CT value less than 0.25 mg/L·min at pH 7, 23°C (42). Furthermore, the NHRMC reports published chlorination LRVs for *E. coli* of 2 log10 achieved with a CT value of <1 mg/L·min at pH 6–10, 10–15°C (5), consistent with findings from this study. No difference was observed in CT values among the 3 pHs (7, 7.5, and 8) or temperatures (10°C and 25°C) tested, likely due to a rapid effect of hypochlorous acid across all experimental regimes tested, resulting in rapid *E. coli* inactivation.

Chloramination is a disinfection process that mixes chlorine with ammonia to form chloramines, which are less reactive but more stable compounds which consequently persist in the water longer than chlorine. Hypochlorous acid reacts rapidly with ammonia to form inorganic chloramines in a series of competing reactions. These competing reactions are primarily dependent on pH and controlled to a large extent by the chlorine-to-nitrogen weight ratio ($Cl_2$:N) (43, 44). Monochloramine is the preferred chloramine species for disinfecting drinking water, as this is more stable than other chlorine species and is predominantly formed in the pH range of 6.5–8.5 with a $Cl_2$:N weight ratio of less than 5:1 (44). Therefore, this study adopted a 4.5:1 ratio for mono-chloramine formation. Free chlorine and ammonia can be applied simultaneously or sequentially in the treatment plant. In this study, the CT values for *in situ* monochloramination (simultaneous addition of chlorine and ammonia) were a magnitude lower (i.e., better disinfection) than the CTs reported for chloramination in the literature (5), but comparable to results generated by chlorination at pH 7 and pH 7.5. Low free chlorine concentrations (<0.1 mg/L) were observed throughout the experiments, suggesting that free chlorine was transiently present in the water and responsible for *E. coli* inactivation before it could form the less bactericidal compound monochloramine. At higher pH (pH 8.0), inactivation was slower and more variable (i.e., larger CTs). However, CTs at pH 8.0 were not as large as published values, indicating that monochloramine was not solely responsible for the inactivation.

As free chlorine likely had a major influence in the *in situ* monochloramination experiments, we also measured the level of inactivation using preformed monochlora-mine. Data obtained utilizing preformed monochloramine demonstrated that CT values increased over two magnitudes (less inactivation) compared to *in situ* monochloramine. These values derived from preformed monochloramine formation were comparable to those reported in the literature. The differences between preformed and *in situ* monochloramine disinfection experiments conducted herein emphasize that the use of this disinfectant at treatment plants may potentially result in greater inactivation rates than credited due to the disinfectant action of hypochlorous acid before chloramine is formed. For both pre-formed monochloramine and *in situ* studies, no significant differences were observed CT values between bloom/non-bloom or encapsulated/non-encapsulated types, suggesting similar inactivation rates for chloramination disinfection. However, it needs to be acknowledged that as in the case of the coagulation-floccula-tion-sedimentation and filtration experimentation, only a limited number of *E. coli* strains were investigated.

UV disinfection is highly effective at inactivating enteric bacteria, *Cryptosporidium*, *Giardia*, and some viruses. In contrast to chlorination and chloramination, no detrimental by-products have been reported in UV-treated effluent (45, 46), making it a choice disinfection method. UV penetrates the cell wall and transfers electromagnetic energy to an organism's genetic material, damaging DNA and limiting the cell's ability to function or reproduce (45, 46). Depending on the dose and wavelengths of UV applied, UV can also damage proteins, which also negatively affects cell function (45, 47). The US EPA (24) reported that the UV dose ranges required to achieve 1, 2, and 3 LRVs are 1.5–4.4, 2.8–6.2, and 4.1–7.3, respectively. In this study, the UV dose necessary to

achieve these LRVs across all strains was almost entirely within these stated ranges. While strain A_E258 was found more resistant to UV inactivation, no significant differences were observed in the UV dose required for inactivation between bloom/non-bloom or encapsulated/nonencapsulated types. As one of the primary targets of UV damage is the cellular DNA, differences in resistance to UV disinfection for *E. coli* are likely to arise based on differences in cellular mechanisms that have evolved to avoid and repair this damage (48–50). While it has been hypothesized that the group 1 capsule may be critical for bloom formation, enhanced survival may not involve such pathways that repair DNA damage. Instead, enhanced survival may be associated with an ability to evade protozoan predation: inferred from phagocytic resistant capsular serotypes (51), an ability of some encapsulated bacteria to attach to surfaces and form resistant biofilms in infected hosts (52–54) or others to exhibit higher thermo-tolerance (55). Moreover, recent work has suggested that the structure of the *Klebsiella* capsule may be more conducive to nutrient uptake or utilization, driving a growth advantage (13). This, together with the protective roles played by the capsule and the shorter lag phase of capsule-positive strains, may explain why capsule-positive strains produce elevated counts in response to nutrient influx.

The majority of the bacterial cells within the environment are predominantly present in stationary phase as the natural habitat of bacteria often contains limited nutrients —so rapid growth is often hampered (56). In stationary phase, bacterial cells become spherical and smaller with a rigid cell envelope, the cell wall is highly cross-linked, membrane fluidity reduces, and cells activate a stringent response mechanism to survive adverse environmental conditions. These changes that occur during stationary phase are a means of bacterial adaptation by which bacteria survive under conditions of stress or starvation (57). Previous work has demonstrated that after starvation *E. coli* passes into stationary phase—affording some protection against harsh environmental conditions (58). The metabolic nature of the *E. coli* ("fresh" or "starved") was, therefore, examined to understand if it affected removal or disinfection behavior between bloom and non-bloom formers or encapsulated vs and non-encapsulated *E. coli*. No differences to treatment removal or disinfection of any of strains after induction of stationary phase were, however, identified. This suggests that *E. coli* strains in the environment—bloom or non-bloom, encapsulated or non-encapsulated, and regardless of metabolic phase—will be susceptible to standard treatment and disinfection processes. Future investigations understanding how capsular expression is affected by environmental stress and growth stage may provide some insight into the importance of the capsules in survival of environmental *E. coli*.

The responses of bloom-forming environmental *E. coli* strains to common water treatment and disinfection processes were assessed and compared to *E. coli* strains from human and bird faeces. The processes evaluated were coagulation (with alum or ferric iron compounds) flocculation-sedimentation (Jar test), filtration, chlorination, chloramination, and UV disinfection. These experiments showed that the responses of bloom isolates were not significantly different from those of fecal *E. coli* strains. Therefore, commonly used water treatment and disinfection processes should be considered to be effective for removing bloom-forming *E. coli* strains from water.

## MATERIALS AND METHODS

### *E. coli* storage

Twenty-eight *E. coli* reference strains received from the Australian National University (ANU) courtesy of Professor David Gordon were streaked onto Tryptone Soy Agar (TSA, Oxoid) plates and incubated at 37°C. Colonies were collected and cryopreserved using Microbank vials (Microbank, ProLab Diagnostics) following the manufacturer's instructions. The Microbank vials were then stored at −80°C until strains were required for experimentation.

## *E. coli* genetic analysis

The *E. coli* reference strains received were characterized using repetitive sequence-based polymerase chain reaction (rep-PCR). This was undertaken with the bioMérieux *E. coli* rep-PCR kit using an "in-house" protocol from the Australian Water Quality Centre (AWQC). Briefly, PCR products were separated by electrophoresis to generate fingerprints, which were analyzed by DiversiLab|bioMérieux software and compared to a proprietary AWQC database. Furthermore, quadruplex PCRs were performed as previously described to assign the isolates to appropriate phylotypes (59–62). All phylogroup determinations using the quadruplex PCRs agreed with the information provided by ANU. Furthermore, clustering derived from the 28 *E. coli* reference strains fingerprinting was overall consistent with the phylogrouping. However, there were a few exceptions where phylogroup designations and fingerprint clusters were not concordant. This was not unexpected as the quadruplex PCRs, and the assignment to phylogroups is based on the phylogenetic analysis derived from four coding genes, unlike clustering derived from the rep-PCR and DiversiLab | bioMérieux software, which is derived from repetitive gene element analysis.

## Strain selection

A total of six strains were selected for experimentation. The rationale behind strain inclusion was to incorporate strains covering both bloom and non-bloom-forming *E. coli*. The *E. coli* phylogroup and presence or absence of a capsule formed were additional criteria for inclusion. The chosen strains comprised three bloom isolates (one Australian East coast strain, one Australian West coast strain, and one capsule-negative variant of a West coast strain) and three isolates of fecal origin (one capsule-negative human strain, one capsule positive human strain, and one capsule-negative bird strain). The selected strains and their characteristics are listed in Table 8. The capsule-free strain was isolated co-incidentally and represented less than 10% of the total strains examined.

## Propagation of exponential phase *E. coli* for disinfection experimentation

Actively growing "fresh" *E. coli* strains were cultured on the day of experimentation for use in disinfection investigations. Briefly, a bead from the Microbank vial was aseptically placed into a 10 mL Tryptone Soy Broth (TSB, Oxoid) and incubated for 3–5 h on an orbital shaker (120 rpm) at 37°C. Cultures were harvested when the optical density of the culture (Genesys 6, Thermo) read between 0.6 and 0.8 at OD600 (Optical Density). This was undertaken to ensure that cultured *E. coli* were actively growing (exponential/log phase). Cultures (5 mL) were then collected and centrifuged at 3,273 *g* (Allegra X-12R centrifuge, Beckman Coulter), at 8°C for 30 min. The supernatant was removed via aspiration before the bacterial pellet was resuspended in phosphate-buffered saline (PBS) and centrifuged at 1,321 *g*, at 8°C for 30 min. This centrifugation/wash step was repeated a further two times. The culture was then resuspended into PBS (5 mL) ready for use in disinfection experiments.

**TABLE 8**  Characteristics of *E. coli* strains selected for experimentation

| Strain designation | Phylogroup | Bloom forming | Capsule |
| --- | --- | --- | --- |
| B1_1330_2_1_WA_Bloom | B1 | Yes | No |
| B1_1863_3_1_WA_Bloom | B1 | Yes | Yes, KL31 |
| A_E258_A1_Bloom_K16 | A | Yes | Yes, KL16 |
| C_H218_non-bloom_KL16 | C | No | Yes, KL16 |
| A_H386_non-bloom | A | No | No |
| B1_B226_non-bloom | B1 | No | No |

## Preparation of "starved" *E. coli* for treatment and disinfection experimentation

*E. coli* were described in the above section but were starved by placing the culture at 4°C for 24 to 26 h, which induces the stationary phase of growth and better reflects the condition of cells in environmental waters (63). These prepared cultures were then used in treatment process and disinfection experiments.

## Water treatment and disinfection methods

Drinking water treatment often consists of a combination of physical removal and disinfection. Each process was considered separately herein for the purposes of experimentation.

## Water treatment (treatment by physical removal)

Commonly used water treatment processes, namely, coagulation/flocculation, sedimentation, and filtration, were investigated to determine their effect on the removal of different *E. coli* strains from source water. Two untreated source waters—Happy Valley Reservoir (South Australia) and Prospect Reservoir (New South Wales)—were used for these studies. Attempts were made to replicate the full-scale water treatment process as close as possible at the laboratory scale (Jar Testing). As the effectiveness of particle removal is dependent on the coagulation conditions employed (i.e., dose, pH, temperature, turbidity, etc.), jar test experiments were performed to evaluate the removal of *E. coli* under existing operating conditions (chemical doses and pH). "Starved" *E. coli* strains were examined to identify if differences in removal behavior existed between bloom and non-bloom formers, capsule and non-capsule *E. coli*, as well as between phylogroups. *E. coli* were enumerated using a membrane filtration technique in combination with chromogenic MI agar incubated for 22–24 h at 36°C. Basic water chemistry tests were undertaken on all waters (pH, turbidity, color, UV absorbance, and dissolved organic carbon) using Australian Standard Methods at the AWQC (64). Experiments were repeated in triplicate.

## Jar testing and filtration to mimic full-scale Happy Valley treatment plant

Fifty liters of Happy Valley Reservoir water was collected from the treatment plant inlet and stored for no longer than a week before use in jar testing experiments. For each experiment, a total of six jars, each containing 2 L of raw water, were inoculated with 1 mL of "starved" *E. coli* (a different strain in each jar). Jar test conditions replicated chemical doses used at the time the water was taken. Each sample was mixed for 5 min at 20 rpm with control samples (10 mL) collected from each jar after 4 min of mixing. Alum (coagulant) was added to a final concentration of 79 mg/L followed by flash mixing for 1 min at 200 rpm. Samples were then flocculated for 14 min at 20 rpm before settling for 15 min. After sedimentation, two samples (120 and 500 mL) were collected for *E. coli* analysis and basic water chemistry analysis. The remaining settled water was then filtered through a Whatman No.1 membrane (folded, Ø 90 mm) using a Urbanti–Bel-Art Filter funnel under gravity. Filtered samples (120 and 500 mL) were subsequently used for *E.* coli analysis and basic water chemistry analysis.

## Jar testing and filtration to mimic full-scale Prospect water treatment plant

Fifty liters of Prospect Reservoir water was collected from the inlet and stored for no longer than 2 weeks before use in jar testing experiments. For each experiment, a total of six jars, each containing 2 L of pH pre-adjusted raw water (lime addition to raise pH from 6 to 8.5), were inoculated with 1 mL of "starved" *E. coli*. Jar test conditions replicated chemical doses used at the time the water was taken. Each sample was mixed for 5 min at 20 rpm with control samples (10 mL) collected from each jar after 4 min of mixing. Ferric chloride (coagulant) was added to a final concentration of 8 mg/L,

followed by a 1-min flash mix at 200 rpm. The flocculation aids, p-DADMAC (1 mg/L) and LT20 (0.1 mg/L), were added immediately after ferric chloride dosing. Samples were then flocculated for 14 min at 20 rpm before settling for 15 min. After sedimentation, two samples (120 and 500 mL) were collected for *E. coli* analysis and basic water chemistry analysis. The remaining settled water was then filtered through a Whatman No.1 membrane. Filtered samples (120 and 500 mL) were subsequently used for *E. coli* analysis and basic water chemistry analysis.

## Disinfection methods (treatment by disinfection)

Glassware used for disinfection experiments was acid washed overnight (5% nitric acid) and rinsed twice with ultra-pure water before oven drying (50℃) to ensure it was demand free (chlorine/chloramine). All disinfection experiments were conducted using buffered demand free (BDF) water. BDF water was prepared by dissolving 0.54 g of anhydrous $Na_2HPO4$ (Sigma) and 0.88 g anhydrous $KH_2PO_4$ (Analar) into 1 L of ultra-pure water (65). The desired pH for disinfection experiments was obtained using 10M NaOH (Sigma) solution and 50% HCl (Sigma) solution. All chlorination/chloramination experiments were conducted in a temperature monitored water-bath (WNE 29, Memmert). *E. coli* were then enumerated by membrane filtration in combination with chromogenic MI agar after a 22–24 h incubation at 36℃. All experiments were repeated in triplicate.

## Chlorine and *in situ* monochloramine CT experiments

Chlorine and *in situ* monochloramine disinfection experiments were undertaken to investigate the effect of the disinfectants on the selected *E. coli* strains. Temperature, pH, and the metabolic nature of the *E. coli* ("fresh" or "starved") were examined to determine if any differences existed in CT inactivation values (the product of the concentration of disinfectant and the contact time) among bloom and non-bloom types, capsule and non-capsule strains, as well as between phylogroup A, B1, and C, under test conditions. The effectiveness of both disinfectants is strongly dependent on pH, temperature, and turbidity. For potable water to meet ADWG (Australian Drinking Water Guideline) requirements, the pH needs to be between 6 and 8.5 (5); therefore, major Australian reticulated supplies try to achieve a pH of 7.5. Hence, pHs 7, 7.5, and 8 were selected for testing under the experimental regime investigated. Water temperatures in major Australian reticulated supplies range from 10℃ to 30℃. However, no guideline values are set for controlling water temperature. As temperatures above 20℃ often result in increased customer complaints (5), it was decided that the highest temperature extremity in this range would not be chosen for testing. Hence, 10℃ (cold) and 25℃ (warm) were selected for use in the experiments conducted herein. Initial disinfection experiments were carried out at pH 7, 7.5, and 8 at 25℃ and pH7.5 at 10℃ with "fresh" *E. coli*. Additionally, the difference between "fresh" and "starved" cultures was also assessed at pH 7.5 at 25℃. All experiments were conducted in a temperature monitored water-bath (WNE29, Memmert). Experiments were repeated in triplicate.

## Chlorine stock concentration and *in situ* monochloramine preparation

A free chlorine stock solution was prepared by bubbling gaseous chlorine through ultra-pure water. Free chlorine concentration was measured using a modification of the standard *N,N*-diethyl-*p*-phenylenediamine-ferrous ammonium sulphate (DPD-FAS) titration method (66), where a one in five dilution of the FAS was performed with a concomitant reduction in volume analyzed. The chlorine stock concentration was measured in triplicate on the day of the test in order to accurately determine the volume required for the desired chlorine dose. All chlorine disinfection experiments were dosed at 1 mg/L free chlorine. Monochloramine was prepared *in situ* by adding chlorine (as described above) and ammonium chloride at a ratio of 4.5:1 concurrently to the sample tested to simulate the treatment process that occurs at the plant. The samples were

vigorously shaken allowing chlorine to be rapidly converted to monochloramine. All experiments were dosed at 1 mg/L *in situ* monochloramine.

## Chlorine and *in situ* monochloramine inactivation

Chlorine and *in situ* monochloramine experiments were carried out separately. For each individual disinfection experiment, a total of six flasks were used, with each flask containing 1.0098 L of pH pre-adjusted BDF inoculated with 1 mL of "fresh"/"starved" *E. coli* (~$10^7$ cfu/100 mL) before incubation in a shaking water bath at the test temperature for a minimum of 2 h prior to disinfection. Control samples (10.8 mL) were collected and neutralized with 1.2 mL of 20% sodium thiosulfate (ST) solution leaving 1L for dosing experiments. In order to determine *E. coli* inactivation by free chlorine, samples (108 mL), at pre-determined time points (0.25, 0.5, 0.75, 1, 1.5, and 2 min) were collected and neutralized with 12 mL of 20% ST. To determine *in situ* monochloramine inactivation of *E. coli*, samples (108 mL) were collected (0.25, 0.5, 0.75, 1, 1.5, and 2 min) before being neutralized with 12 mL of 20% ST. *E. coli* were then enumerated by membrane filtration in combination with chromogenic MI agar incubated for 22–24 h at 36°C. The pH of each experimental flask was determined at both the start and finish of each experiment to ensure that the disinfection experiments were carried out at the desired pH (±0.1) and temperature ±0.5°C throughout the duration of the experiment.

## CT calculation for chlorine and *in situ* monochloramine disinfection

To determine experimental CT value, samples (20 mL) were taken from test flasks at time points 0.5, 1, 1.5, and 2 min. The free available chlorine (FAC), monochloramine, along with other chlorine species was measured immediately using the DPD-FAS method. As the experiments were conducted in demand-free water, chlorine decay was not expected. Hence, the dosed concentration was determined by the average readings of FAC/monochloramine measured from the experiments. The CT values at the measured time points were calculated by multiplying the average FAC/monochloramine dosed concentration with the time of exposure. CTs to achieve a 1, 2, and 3 $log_{10}$ inactivation were calculated from the rising phase of the graph of *E. coli* inactivation (excluding the plateau) using either a linear regression or polynomial equation of best fit.

## Preformed monochloramine CT experiment

Preformed monochloramine disinfection experiments were undertaken to investigate the effect of the disinfectant on the selected *E. coli* strains. Chloramination is normally performed at pH 8.5 ± 0.2. Like chlorination, it is known that the CT requirements are reduced at higher temperature (3). Hence, pH 8.5 at 25°C was selected for experimentation. "Starved" *E. coli* were examined to determine if CT values for inactivation differed between bloom and non-bloom types, capsule and non-capsule strains and between phylogroups under test conditions. All experiments were conducted in a temperature monitored water-bath (WNE 29, Memmert). Preformed monochloramine was prepared by reacting chlorine solution and ammonium chloride at a ratio of 4.5:1. Monochloramine has an optimum stability at pH 9. Hence, the solution was pH-adjusted to 9 using sodium hydroxide. The preformed monochloramine was stored in a dark bottle for a maximum 2 weeks with minimal deterioration. All *E. coli* inactivation experiments were dosed at 1 mg/L preformed monochloramine.

## Preformed monochloramine inactivation

A total of six flasks containing 1.16 L of pH pre-adjusted BDF were inoculated with 1.15 mL of "starved" *E. coli* (~$10^{5-6}$ cfu/ 100 mL) before incubation in a shaking water bath at the test temperature for a minimum of 2 h prior to disinfection. Control samples (10.8 mL) were collected and neutralized with 1.2 mL of 20% ST solution prior to preformed monochloramine dosing. To determine *E. coli* inactivation by monochloramine, samples (108 mL) were taken at pre-determined time points (1, 5, 10, 15, 20,

25, 30, and 60 min) and neutralized with 12 mL of a 20% ST solution. Samples were then enumerated by membrane filtration in combination with chromogenic MI agar incubated for 22–24 h at 36°C. The pH of each experimental flask was determined at both the start and finish of each experiment to ensure that the disinfection experiments were carried out at the desired pH (±0.1) and temperature ±0.5°C throughout the duration of the experiment.

## CT calculation preformed monochloramine disinfection

To determine experimental CT value, samples (20 mL) were taken from test flasks at time points 1, 5, 10, 15, 20, 25, 30, and 60 min. The monochloramine (along with other chlorine species) was measured immediately using the DPD-FAS method. As the experiments were conducted in demand-free water, chloramine decay was not expected. Hence, the dosed concentration was determined by the average readings of monochloramine measured from the experiments. The CT value at the measured time points was calculated by multiplying the average monochloramine dosed concentration with the time of exposure CTs to achieve a 1, 2, and 3 log10 inactivation were calculated from the rising phase of the graph of *E. coli* inactivation (excluding the plateau) using either a linear regression or polynomial equation of best fit.

## Low-pressure UV disinfection and exposure time calculation

A bench-scale collimated beam apparatus (Trojan Technologies Inc., Ontario, Canada) was used to irradiate *E. coli* ("fresh" and "starved") to determine if any differences in UV inactivation exist among bloom and non-bloom types, capsule and non-capsule strains, as well as between phylogroup A, B1, and C were discernible under test conditions. This apparatus contained a low pressure 254 nm mercury lamp. Experiments were performed at room temperature (approximately 22°C) in triplicate.

On the day of experimentation, irradiance was measured with a radiometer (International Light, Model 1L1400A, equipped with a 254 nm UV detector model no. XRL140T254, Newburyport MA) calibrated to the standards of the US National Institute of Standards and Technology (NIST). The low-pressure UV doses were determined as previously described (67) using a Microsoft Excel spreadsheet provided by the authors (Bolton Photosciences, Ayr, Canada). A petri factor, reflection factor, and water factor were applied to all calculations.

## UV inactivation experiments

A total of six flasks containing 1.2 L of BDF water at pH 7.5 were inoculated with 1 mL of "fresh"/"starved" *E. coli* (~$10^7$ cfu/100 mL). A 160 mL of sub-sample was exposed to varying UV doses (1, 3, 5, 7, 10, 15, and 20 mJ/cm$^2$) using a collimated beam bench top test unit to develop a dose-response curve. After UV exposure, the samples were covered with aluminum foil as *E. coli* can undergo photo repair. The remaining sample (unexposed) was collected and used as the non-irradiated control. The concentration of *E. coli* was enumerated by the membrane filtration technique in combination with chromogenic MI agar incubated for 22–24 h at 36°C.

## Statistical analysis

The differences between strains were analyzed using one-way analysis of variance (ANOVA), Tukey's multiple comparison test (GraphPad Prism 5, USA). The comparison between "fresh" vs "starved", bloom vs non-bloom, and capsule vs non-capsule was analyzed using nonparametric Mann-Whitney *t*-tests, two tailed (GraphPad Prism 5, USA). Statistical significance was defined as $P$ value ≤ 0.05. Non significance was defined as $P$ value > 0.05.

## ACKNOWLEDGMENTS

We thank Con Kapralos, Renae Phillips, David Cook, Lisa Teakle and Rolando Fabris of SA Water for their technical assistance and advice. We are grateful to Professor David Gordon from the Australian National University for provision of the *E. coli* strains used in this study.

This work was funded by Water Research Australia WRA Project 1101, Management of environmental *E. coli*.

M.L. (Methodology, Formal analysis, Investigation, Writing - Review & Editing), P.M. (Conceptualization, Methodology, Formal analysis, Writing - Review & Editing, Funding acquisition), B.K. (Writing - Original Draft, Writing - Review & Editing, Visualization, Supervision, Project administration).

## AUTHOR AFFILIATION

[1]South Australian Water Corporation, Adelaide, South Australia, Australia

## AUTHOR ORCIDs

Brendon J. King  http://orcid.org/0000-0002-6114-0950

## FUNDING

| Funder | Grant(s) | Author(s) |
| --- | --- | --- |
| Water Research Australia (WaterRA) | | Paul T. Monis |
| | | Brendon J. King |

## DATA AVAILABILITY

All data supporting this study is presented within this article.

## ADDITIONAL FILES

The following material is available online.

### Supplemental Material

**Supplemental material (Spectrum00856-24-s0001.pdf).** Tables S1 and S2.

### Open Peer Review

**PEER REVIEW HISTORY (review-history.pdf).** An accounting of the reviewer comments and feedback.

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
