## [Reviewer comments · Microbiology Spectrum]

Microbiology Spectrum

The Efficacy of Current Treatment Processes to Remove, Inactivate or Reduce Environmental Bloom Forming *Escherichia. coli*

Melody Lau, Paul Monis, and Brendon King

Corresponding Author(s): Brendon King, South Australian Water Corporation

Review Timeline:

Submission Date:	April 3, 2024
Editorial Decision:	May 28, 2024
Revision Received:	May 31, 2024
Accepted:	June 6, 2024

Editor: Blaire Steven

Reviewer(s): Disclosure of reviewer identity is with reference to reviewer comments included in decision letter(s). The following individuals involved in review of your submission have agreed to reveal their identity: GRACE ONYUKWO ABAKPA (Reviewer #1)

Transaction Report:

DOI: <https://doi.org/10.1128/spectrum.00856-24>

Re: Spectrum00856-24 (The Efficacy of Current Treatment Processes to Remove, Inactivate or Reduce Environmental Bloom Forming *Escherichia. coli*)

Dear Dr. Brendon King:

I apologize for the delay in review. I had difficulty securing a second reviewer. After reading the paper myself I am pleased to inform you that your manuscript has been editorially accepted for publication. However, there are a few additional questions in the submission form that need to be answered before the final decision. Once these are completed, please return your submission so that I can move your paper forward to acceptance.

Thank you for the privilege of reviewing your work. Below you will find my comments, instructions from the Spectrum editorial office, and the reviewer comments.

Revision Guidelines

Sincerely,
Blair Steven
Editor
Microbiology Spectrum

Reviewer #1 (Comments for the Author):

The paper adds to knowledge but effect stated out corrections.

Editor comments:

Review comments

1. Line 74: The word aims should be corrected to aim.
You have a specific aim and can have multiple objectives.
2. Line 254: Ferric??????
3. Line 257: For two waters(Wrong English)
4. Rephrase
5. Line 307: State the reference for the previously published data.
6. Line 309: Ref for NHRMC
7. Line 499 change 'is' to was.

Review comments

1. Line 74: The word aims should be corrected to aim. You have a specific aim and can have multiple objectives. We have rephrased lines 74-76 to ensure a singular aim with the two objectives as indicated by the referee.

2. Line 254: Ferric????? Now line 254. Changed to ferric coagulants

3. Line 257: For two waters(Wrong English) This has been rephrased to “for both waters—Happy Valley and Prospect—....

4. Rephrase See above

5. Line 307: State the reference for the previously published data. We do with examples outlined from lines 308 to 312.

6. Line 309: Ref for NHRMC We provided this as reference 5 on line 312.

7. Line 499 change ‘is’ to was

The subject of the sentence is “The effectiveness of both disinfectants”, is singular, so the singular verb “is” should be used. Now line 501.

Re: Spectrum00856-24R1 (The Efficacy of Current Treatment Processes to Remove, Inactivate or Reduce Environmental Bloom Forming *Escherichia. coli*)

Dear Dr. Brendon King:

Thank you for your patience with the review process.

Your manuscript has been accepted, and I am forwarding it to the ASM production staff for publication. Your paper will first be checked to make sure all elements meet the technical requirements. ASM staff will contact you if anything needs to be revised before copyediting and production can begin. Otherwise, you will be notified when your proofs are ready to be viewed.

Sincerely,
Blair Steven
Editor
Microbiology Spectrum